# GLI3 Is Required for OLIG2+ Progeny Production in Adult Dorsal Neural Stem Cells

**DOI:** 10.3390/cells11020218

**Published:** 2022-01-10

**Authors:** Rebecca J. Embalabala, Asa A. Brockman, Amanda R. Jurewicz, Jennifer A. Kong, Kaitlyn Ryan, Cristina D. Guinto, Arturo Álvarez-Buylla, Chin Chiang, Rebecca A. Ihrie

**Affiliations:** 1Department of Cell & Developmental Biology, Vanderbilt University School of Medicine, Nashville, TN 37232, USA; rebecca.j.embalabala@vanderbilt.edu (R.J.E.); asa.a.brockman@vanderbilt.edu (A.A.B.); amanda.jurewicz@nyulangone.org (A.R.J.); jennifer.a.kong@vanderbilt.edu (J.A.K.); katieryan5@gmail.com (K.R.); chin.chiang@vanderbilt.edu (C.C.); 2Department of Neurological Surgery, University of California—San Francisco, San Francisco, CA 94143, USA; cristina.guinto@ucsf.edu (C.D.G.); abuylla@stemcell.ucsf.edu (A.Á.-B.); 3Eli and Edythe Broad Institute of Regeneration Medicine, University of California—San Francisco, San Francisco, CA 94143, USA; 4Department of Neurological Surgery, Vanderbilt University School of Medicine, Nashville, TN 37232, USA

**Keywords:** ventricular–subventricular zone (V-SVZ), neurogenesis, Sonic hedgehog (Shh), oligodendrocytes, Gli3, radial glial cells, Smoothened (SMO), adult neural stem cells, gliogenesis

## Abstract

The ventricular–subventricular zone (V-SVZ) is a postnatal germinal niche. It holds a large population of neural stem cells (NSCs) that generate neurons and oligodendrocytes for the olfactory bulb and (primarily) the corpus callosum, respectively. These NSCs are heterogeneous and generate different types of neurons depending on their location. Positional identity among NSCs is thought to be controlled in part by intrinsic pathways. However, extrinsic cell signaling through the secreted ligand Sonic hedgehog (Shh) is essential for neurogenesis in both the dorsal and ventral V-SVZ. Here we used a genetic approach to investigate the role of the transcription factors GLI2 and GLI3 in the proliferation and cell fate of dorsal and ventral V-SVZ NSCs. We find that while GLI3 is expressed in stem cell cultures from both dorsal and ventral V-SVZ, the repressor form of GLI3 is more abundant in dorsal V-SVZ. Despite this high dorsal expression and the requirement for other Shh pathway members, GLI3 loss affects the generation of ventrally-, but not dorsally-derived olfactory interneurons in vivo and does not affect trilineage differentiation in vitro. However, loss of GLI3 in the adult dorsal V-SVZ in vivo results in decreased numbers of OLIG2-expressing progeny, indicating a role in gliogenesis.

## 1. Introduction

The ventricular–subventricular zone (V-SVZ) is one of two known sources of new neurons in the adult mammalian brain. Glial fibrillary acidic protein (GFAP)-expressing neural stem cells (NSCs) in the V-SVZ give rise to transit-amplifying cells (also known as C cells) which then generate immature glial progeny and neuroblasts [1]. V-SVZ-derived neuroblasts migrate to the olfactory bulb, via the rostral migratory stream, and differentiate into distinct cell types in the granular and periglomerular layers. Prior models assumed that neural stem cells in the V-SVZ were homogeneous and able to generate multiple neuronal cell types. However, more recent research using viral or genetic lineage tracing has determined that NSCs are heterogeneous and depending on their location generate different types of neurons [2,3,4,5,6,7]. The dorsal V-SVZ primarily generates superficial granule interneurons, a subset of periglomerular cells, and oligodendrocytes. The ventral V-SVZ generates deep granule interneurons and calbindin-positive periglomerular cells [5,8]. The adult V-SVZ is considered a mosaic of NSCs with positional information that is encoded during early embryonic development and maintained in adulthood at least partially through epigenetic regulation [9,10]. Since the discovery of NSC positional identity, many studies have focused on identifying mechanisms that are required in this patterning and its maintenance.

Sonic Hedgehog (SHH) is a secreted ligand that is important in NSC activity at multiple stages in development. When SHH ligand is bound to the receptor Patched (PTC), SHH signaling is transduced through the now derepressed Smoothened (SMO) coreceptor [11,12]. After a series of incompletely characterized steps in signal transduction, the Glioma associated oncogene transcription factors (GLIs) GLI2 and GLI3 are modified [13,14]. At high levels of activity, the transcriptional activator GLI2 causes transcription of GLI1, which is itself a constitutive transcriptional activator. However, without the SHH ligand, GLI3 is present in its cleaved form as a transcriptional repressor (GLI3R), which inhibits the transcription of ventral identity genes. GLI2 has also been postulated to have a repressor form with similar properties. *Gli1* transcript is elevated in the ventral aspect of the V-SVZ, while *Gli3* is expressed at higher levels in the dorsal V-SVZ and *Gli2* is present in both subregions [15,16].

Studies of the Shh signaling pathway have identified multiple intersecting functions for different pathway members, with some proteins broadly required for neurogenic activity across the niche and others being dispensable within specific subregions or populations. Models used to study the Shh signaling pathway in vitro and in vivo include conditional alleles of *Gli2*, *Gli3*, *Ptc*, and *Smo*, as well as reporter alleles driven by the *Gli1* or *Shh* locus [15,16,17,18,19,20]. When lacZ knock-in alleles were used to visualize GLI factor expression, all three GLI factors were found in the adult V-SVZ with GLI1 found primarily in the ventral region [16,17,21]. SHH signaling has been suggested to be required for the patterning that produces ventral neuronal progeny, especially the specification of adult NSCs in the V-SVZ [16]. However, more recent studies indicate that once specification occurs, high SHH signaling is dispensable for regional specification of ventral neurogenic progenitors in juvenile or adult V-SVZ [9]. Genetic ablation of the SMO coreceptor driven by NKX2.1, a GLI target gene expressed in ventral subregions, did not impact the expression of transcriptional markers of ventral V-SVZ neural NSCs. Prior work demonstrated that ablating SMO in V-SVZ cells led to a decrease in neurogenesis, though the removal of GLI2 or GLI3 in GFAP-expressing cells resulted in no significant change in V-SVZ proliferation or OB interneuron formation [15]. Selective loss of GLI2 and GLI3, in mice with concomitant loss of function of SMO, indicate that the repressor functions of both GLI2 and GLI3 may contribute to the reduction in neurogenesis observed after SMO loss in the postnatal V-SVZ.

The dorsal region of the V-SVZ is responsible for generating oligodendrocytes in early postnatal life, as intersectional lineage tracing revealed that dorsal radial glial cells that transiently express *gli1* during the first postnatal week produce oligodendroglial cells. When SMO is inactivated, oligodendrocyte production decreases, and induction of SMO activity results in increased oligodendrogenesis [8]. In contrast, ventral NSCs do not generate large numbers of oligodendrocyte precursor cells or mature oligodendrocytes [22]. However, in conditions of injury where there has been a loss of mature oligodendrocytes, GLI1 has been associated with the differentiation of ventral NSCs into oligodendrocytes [23,24]. In the absence of GLI1 following demyelination, GLI2 is required for this differentiation of ventral NSCs into oligodendrocytes [22]. While the above suggests that GLI1 and GLI2 are necessary for the generation of glial cells from the adult V-SVZ, the effects of loss of GLI3, and the consequent modulation of SHH signaling, on the dorsal production of oligodendrocytes have not been studied. Here, we use focal ablation of GLI3 and GLI2 in subregions of the adult SVZ to test the requirement for these proteins in neuronal and oligodendrocyte production. We find that interneuron subtype formation in the olfactory bulb is not altered when GLI3 is ablated singly or in concert with GLI2, although effects on total numbers of interneurons are seen in the ventral subregion, and that loss of GLI3 in cultured perinatal neural progenitors from either region does not affect production of neurons or glia. However, loss of GLI3 in the adult dorsal V-SVZ does reduce the number of OLIG2+ cells, indicating an ongoing role for regulation of SHH pathway members in gliogenesis in this subregion.

## 2. Methods

### 2.1. Protein Extraction and Western Blotting

Cultured cells derived from dorsal or ventral V-SVZ of wild type mice (P2) were harvested for extracts for Western blot analysis. Cells were lysed using a radioimmunoprecipitation assay (RIPA) buffer containing 50 mM Tris (at pH 7.5), 150 mM NaCl, 1 mM EDTA, 1% Triton X-100, 0.1% SDS, and 1% sodium deoxycholate. Phosphatase inhibitor cocktails I and II (Sigma Aldrich, St Louis, MO, USA: P-2850 and P-5726, respectively), Complete protease inhibitor cocktail (Roche, Indianapolis, IN, USA), and 1 mM phenylmethylsulfonyl fluoride (Sigma Aldrich, St Louis, MO, USA, Cat #P7626) were added to the tissue samples suspended in buffer, which were then allowed to rest on ice for 30 min. After incubation, samples were centrifuged at 16,000× *g* for 20 min at 4 °C to remove insoluble debris. Concentration of lysate samples was determined using a bicinchoninic acid (BCA) kit (Pierce, Waltham, MA, USA). Lysates were split into 30 μg aliquots and separated on a 4 to 10% Tris-glycine SDS-PAGE with Precision Plus protein standards (Bio-Rad, Hercules, CA, USA) and transferred to nitrocellulose membranes. Membranes were incubated with anti-Gli3 antibody (6F5 from Genentech, South San Francisco, CA, USA) at 2–5 μg/mL concentration in 5% milk in Tris-buffered saline plus 0.05% Tween 20. Secondary antibodies used were conjugated with horseradish peroxidase (HRP), and bands were visualized with ECL substrate on film. Quantification of individual films was performed using the integrated density function in ImageJ. Specificity of the 6F5 antibody was confirmed on each blot by sizing and by inclusion of a positive control with known patterns of GLI3 expression (3T3 cells with and without Smoothened agonist).

### 2.2. Animal Lines

*Smo^FLX/FLX^*, *Gli2^FLX/FLX^*, *Gli3^FLX/FLX^*, *CAG*, and *Ai14* lines used here have been previously described [25,26,27,28,29] and were obtained from Jackson Laboratories (Bar Harbor, ME, USA). Loss of the relevant transcript/protein when Cre was introduced was confirmed via qRT-PCR and Western blotting in cultured cells. Both male and female mice were used in all experiments.

### 2.3. Stereotaxic Injections

For focal ablation of SMO, GLI2, and/or GLI3 in adult (P30–60) animals, 50 nL of Ad:GFAPp-Cre (approximately 10^9^ viral particles, prepared as in [5]) was injected at dorsal coordinates (0.5 mm A, 3.2 mm L to bregma, 1.8 mm from the pial surface, with needle at 45 degrees) or ventral coordinates (0.5 mm A, 4.6 mm L to bregma, 3.3 mm from the pial surface, with needle at 45 degrees) using a beveled pulled glass micropipette needle. Control injections with Ad:CMV-GFP (Vector Biolabs, Malvern, PA, USA, Cat #1060) were conducted using the same coordinates. At 30 days after viral injection, a subset of animals was injected with 50 mg/kg bromodeoxyuridine 1 h prior to being transcardially perfused with 0.9% saline followed by 4% paraformaldehyde in 0.1 M sodium phosphate buffer. Dissected brains were postfixed overnight in 4% PFA, and prepared as 50-micron floating sections for immunostaining.

### 2.4. V-SVZ Dissection and Tissue Culture

Animal procedures were performed according to approved IACUC protocol. Postnatal day 2 pups were rapidly decapitated with sterile fine-point dissecting scissors and the heads were cleaned sequentially using betadine and 100% ethanol. Brain regions were microdissected on an aseptic tray under a dissecting microscope. A cross section of approximately 2–4 mm over the region containing the V-SVZ and the lateral ventricles was taken using a razor blade. Dorsal and ventral regions of the V-SVZ were removed using fine point forceps and a microsurgical knife. Collected tissue was placed in 1.5 mL Eppendorf tubes containing DMEM/F12+Glutamax (ThermoFisher, Waltham, MA, USA, Cat #10565018), and kept on ice until tissue from pups in an entire litter was aggregated. Once tissue was collected, it was spun down at a low speed until a pellet formed. Once macerated with a sterile microsurgical knife, tissue was resuspended in 0.25% trypsin, and incubated at 37 °C with 5% CO_2_ for 20 min on a nutating mixer. After 20 min, trypsin was quenched by adding FBS and DMEM/F12+Glutamax. Tissue was then triturated using a P1000 pipette to form a single-cell suspension, washed with media, and centrifuged for 5 min at 300× *g* to form a pellet which was resuspended in N5 media for cell culture.

After 2 passages in culture, dorsal or ventral progenitor cells were plated on 8-well chamber slides which had previously been coated with 0.1 mg/mL poly-D-lysine hydrobromide (Sigma Aldrich, St Louis, MO, USA, Cat# P6407) and 0.02 mg/mL laminin (MilliporeSigma, St Louis, MO, USA, Cat# CC095). After 3–5 days of culture (until cells appeared hyperconfluent) in N5 media, cells were switched to N6 medium containing: 5% fetal bovine serum ([FBS] Corning, Corning, NY, USA, Cat# 35-016-CV), N2 supplement (ThermoFisher, Waltham, MA, USA, Cat# 17502-048, 35 μg/mL bovine pituitary extract ([BPE], ThermoFisher, Waltham, MA, USA, Cat# 13028014), in DMEM/F12+Glutamax (as in [9]).

### 2.5. Viral Transduction In Vitro

Ad-CMV-Cre (Vector BioLabs, Malvern, PA, USA, Cat# 1045) was added at a concentration of 1 × 10^7^ pfu/mL to N5 media containing: 5% fetal bovine serum, N2 supplement, 35 μg/mL bovine pituitary extract, 20 ng/mL recombinant mouse basic fibroblast growth factor (bFGF, ThermoFisher, Waltham, MA, USA, Cat# PMG0035), 20 ng/mL epidermal growth factor (EGF, ThermoFisher, Waltham, MA, USA Cat. #53003-018), in DMEM/F12+Glutamax. Media for viral transduction was allowed to incubate with cells for 48 h at 37 °C, 5% CO_2_. Cell culture expression of tdTomato was verified using a fluorescence capable inverted microscope.

### 2.6. Immunofluorescence Analyses

Brain sections were stained as floating sections prior to mounting onto plus coated slides (ThermoFisher, Waltham, MA, USA Cat. # 12-550-19). For staining, sections were blocked for 30 min using 1% normal donkey serum (Sigma Aldrich, St Louis, MO, USA, Cat#D9663), plus 0.1% Triton X-100 (Sigma Aldrich, St Louis, MO, USA, Cat#X100) in 1X PBS at room temperature. Primary antibodies were diluted in blocking buffer and added to sections that were allowed to incubate overnight with gentle agitation at 4 °C. After incubation, tissue was rinsed with 1X PBS (without Ca^2+^/Mg^2+^) three times. Sections were incubated for 1 h at room temperature in secondary antibodies diluted in blocking buffer. Tissue sections were visualized on a Leica SP5 confocal microscope or a Zeiss LSM710 confocal microscope.

For analyses of differentiation in vitro, after 14 days of propagation in differentiation medium, cells were gently rinsed with PBS and fixed with 4% paraformaldehyde in 0.1 M pH 7.4 phosphate buffer for 30 min. Fixed samples were stained according to standard protocols. In brief, cells were permeabilized with 0.2% Triton X-100, followed by staining with primary and secondary antibodies diluted in blocking buffer (2% normal donkey serum in PBS). Antibodies used are detailed below in Table 1. Cells were visualized on a Zeiss LSM710 confocal microscope (all V-SVZ and cultured cell images) or a Nikon Az100 widefield microscope (olfactory bulb slices).

### 2.7. Image Processing and Statistical Analysis

For cultured cell percent positive scoring, nuclei were segmented in CellProfiler [30] and cytoplasmic regions surrounding nuclei were delineated using a defined pixel expansion radius. Nuclear and cytoplasmic marker expressions were measured on each segmented cell. Thresholds for positive marker expression were set per marker by concurrence between two immunostaining experts. At least 180 cells were counted per independent replicate for each condition.

For percent positive scoring of cells in tissue sections, nuclei were segmented using a custom trained Stardist model. Cytoplasmic regions surrounding nuclei were delineated using a defined pixel expansion radius. Nuclear and cytoplasmic marker expressions were measured on each segmented cell. Thresholds for positive marker expression were set by concurrence between two immunostaining experts. Only cells determined to be RFP+ were further scored for markers of interest. Percent positive scores are reported within the RFP+ fraction of cells. Data were plotted per mouse as well as in the aggregate to avoid the masking of any sex- or line-specific differences.

Statistical analyses were performed using GraphPad Prism 9.3.0 (GraphPad, San Diego, CA, USA). *p* values and statistical tests, as well as number of biological/technical replicates, are indicated in figure legends.

### 2.8. Antibodies

Antibodies used for staining are detailed below (Table 1). Each antibody was validated and titrated on known positive and negative controls (cell lines with known transcriptional and protein profiles and tissue sections with known cell compositions). 

## 3. Results

While previous studies have indicated that *Gli3* transcript is present in both dorsal and ventral V-SVZ [16], the relative levels of repressor (cleaved) and full-length forms of the protein have not been reported. We therefore measured the presence of both protein forms in cultured neonatal (P2) progenitor cells via Western blotting (Figure 1). Both dorsal and ventral cells expressed the full-length form of GLI3 (170 kDa), while dorsal cells had higher levels of the shorter GLI3R form (83 kDa). As expected, addition of the small molecule Smoothened agonist (SAG, 10 nM or 100 nM [Selleck Chemicals, Cat #S7779, Houston, TX, USA]) resulted in decreased levels of GLI3R in all cultures, while addition of 250 or 500 nM cyclopamine conversely resulted in decreased levels of the full-length GLI3 (data not shown).

Removal of the SMO coreceptor prevents activation of the canonical SHH signaling pathway, resulting in high levels of GLI3R and pathway repression. To test the requirement for this signaling in the V-SVZ, we ablated SMO expression, and induced GFP reporter expression, in dorsal or ventral V-SVZ using localized injections of Ad:GFAPp-Cre in mice bearing conditional alleles of *Smo* (Figure 2A–H) [26]. In either dorsal or ventral V-SVZ, ablation of SMO resulted in a significant decrease in the total number of labeled interneurons in the olfactory bulb detected at 30 days post injection (Figure 2I,J). This difference was not seen upon injection of a control, non-Cre-expressing virus (Ad:CMVp-GFP) (Figure 2K). Consistent with this finding, we observed fewer GFP-labeled cells at the injection site within the V-SVZ co-labeled with BrdU at 30 days after injection, indicating a smaller number of S phase cells were present. However, this decrease was only reproducibly detectable in the ventral V-SVZ (Figure 2L).

Given the elevated levels of GLI3R in cultured dorsal cells, we next tested the requirement for GLI3 in the generation of dorsally-derived interneurons by focally ablating GLI3 in GFAP-positive type B cells of the dorsal V-SVZ (Figure 3). Following injection of Ad:GFAPp-Cre in *Gli3^FLX/FLX^*; *CAG* adult animals or *Gli3^+/+^*; *CAG* controls [32], virally transduced type B1 cells and their progeny were labeled with GFP in all animals (Figure 3A–F). In control animals, quantification of the location of GFP-labeled cells in the olfactory bulbs at 30 days post transduction revealed the expected distribution of interneurons, with a majority of labeled progeny (approximately 60%) in the superficial granular layer and a minority (~7%) in the deep granule layer (Figure 3J). When GLI3 protein was also lost upon transduction, no differences in the distribution of progeny in the granular layer were seen, consistent with previous reports [15]. Labeling of S phase cells by administration of BrdU one hour prior to sacrifice (30 days after Ad injection) did not reveal differences in rapidly cycling cells within the GFP-labeled population at the injection site when GLI3 was lost (Figure 3G–I). As a second measure of V-SVZ output and comparison to SMO ablation, the number of granular layer interneurons per slice was also quantified (Figure 3Q). Ablation of GLI3 resulted in a small, but not statistically significant (*p* = 0.104, unpaired *t*-test), increase in this number, and *Gli3^FLX/FLX^*; *CAG* animals had slightly more cells per slice than *Gli2^FLX/FLX^*; *Gli3^FLX/FLX^*; *CAG* double mutants (*p* = 0.0034, unpaired *t*-test). However, focal dorsal ablation of both GLI2 and GLI3 did not result in altered distribution of interneurons in the olfactory bulb, suggesting that cell fate was not affected among those neurons generated from the transduced GFAP+ progenitors (Figure 3R). Strikingly, ablation of GLI3 either alone or together with GLI2 in the ventral V-SVZ strongly affected the generation of olfactory interneurons, with a magnitude that was similar to the loss of Smoothened (Figure 3Q).

Progenitors of the V-SVZ, in addition to olfactory bulb interneurons, also generate oligodendrocytes, both during normal development and in adulthood after injury. SHH signaling has also been implicated more broadly in astrocyte production and activation [33]. To examine the production of glia by dorsal and ventral progenitor cells, cultures were prepared from each subregion in postnatal day 2 *Gli3^FLX/FLX^*; *Ai14* animals as well as *Ai14*-only controls. Prior to in vitro differentiation by reduction in growth factors, cultured progenitors were transduced with Ad:CMV-Cre to induce loss of the GLI3 protein and expression of tdTomato. Following 14 days of differentiation, marker expression for neuronal, oligodendroglial and astroglial differentiation was quantified and compared between tdTomato-positive cells and uninfected control cells within the same well, as well as *Ai14*-only controls (Figure 4 and data not shown). All cultures yielded TUJ1-positive neurons, OLIG2+ oligodendrocyte lineage cells, and GFAP-positive astrocytes. A higher fraction of OLIG2+ cells (37%) were seen in dorsally derived cultures (Figure 4A,D,G) relative to matched ventral counterparts (3.7%, *p* = 0.03, unpaired *t*-test)); however, the proportion of cells which were OLIG2-positive did not change when GLI3 was lost (30.4% dorsal, 3.7% ventral). Similarly, no differences were seen between corresponding wild-type cultures without Cre (30% OLIG2+ in dorsal, 1% in ventral, data not shown). GFAP-positive and TUJ1-positive cells were also present in all cultures and did not vary by genotype (GFAP: Figure 4C,F,G; TUJ1: Figure 4B,E,G). As oligodendrogenesis continues in the postnatal period in vivo, and may not be fully reflected by neonatal progenitor cultures, we also analyzed the effects of GLI3 ablation on production of OLIG2+ cells in the adult dorsal V-SVZ. *Gli3^FLX/FLX^*; *Ai14* animals or *Ai14*-only controls were injected in the dorsal V-SVZ with Ad:GFAPp-Cre, and the percentage of RFP-labeled cells expressing OLIG2 was quantified 30 days after injection (Figure 4H–J). Surprisingly, while many OLIG2+ labeled progeny were found in control animals (40% of RFP+ cells at the injection site), this percentage was substantially decreased in *Gli3^FLX/FLX^*; *Ai14* animals (12%), indicating that GLI3 is required for functional postnatal oligodendrogenesis from dorsal V-SVZ.

## 4. Conclusions

How and when positional identity is first encoded in the cells of the V-SVZ remains an active area of investigation, as well as the mechanisms that lead to neuronal and glial lineage generation in different subdomains throughout life. SHH signaling, and specific downstream pathway members, have been implicated in identity specification, stem cell quiescence, and persistence of stem cells during postnatal life [13,19,34]. Pathway inhibition, via generation of GLI3R, is a key step in regulation of this signaling; although, activation by GLI3A has also been suggested to be required in the regional specification of brain structures such as the hypothalamus [35]. The higher abundance of GLI3R protein in the cells of the dorsal V-SVZ suggests that these NSCs may have a higher threshold for activation of SHH signaling, consistent with prior experiments in which short term administration of SHH ligand or SAG in vivo or ex vivo was not sufficient to increase *Gli1*-driven reporter expression [16].

Ablation of the canonical coreceptor SMO within the cells of the V-SVZ results in unattenuated GLI2/3 repressor forms, and thus inhibition of SHH signaling. Prior studies of SMO ablation throughout the V-SVZ found a substantial decrease in the number of labeled OB neurons and in the number of actively proliferating cells within the niche [15]. Using focal viral ablation of SMO, we observed a similar decrease in the total number of labeled progeny in the olfactory bulb, irrespective of dorsal or ventral origin. This confirms that SHH signaling in the postnatal niche likely acts to attenuate repressor activity via GLI2/GLI3, and that this attenuation may affect neural stem cell quiescence or cycling. Consistent with this finding, BrdU incorporation was decreased in SMO-deficient, GFP-positive cells at the site of injection in the ventral V-SVZ, and the concomitant loss of both GLI2 and GLI3, or of GLI3 alone, resulted in a decrease in neuron production by ventral V-SVZ cells only. Surprisingly, a statistically significant difference was not evident in equivalent dorsally-injected animals, suggesting that the activation or persistence of ventral B1 cells is more strongly dependent on active SHH signaling, consistent with prior work highlighting a selective requirement for primary cilia, a key component of canonical SHH signaling, in this region [36].

In early neural tube development, SHH activity and relief from GLI3-mediated repression are required for the specification of multiple neuronal and glial fates, including generation of oligodendrocytes and establishment of the NKX2.1-positive ventral domain via GLI-mediated transcriptional activation [37,38]. In the forebrain, expression of SHH at approximately E10 is required for establishment of the NKX2.1-positive medial ganglionic eminence, which ultimately contributes to a small ventral subdomain of the postnatal V-SVZ [4,39]. The data shown here indicate that when GLI3 protein is removed in the postnatal dorsal V-SVZ, with or without concomitant GLI2 loss, the location of labeled olfactory bulb granule neuron progeny does not change, suggesting that ongoing inhibition of SHH signaling is not required for dorsal neuronal identity in the postnatal V-SVZ. Consistent with these findings and other prior work, ablation of SMO from the V-SVZ at timepoints after identity is established had no effect on the number of NKX2.1-positive progenitors [9]. Collectively, these data argue that after initial patterning events during embryogenesis, NSC neuronal progeny positional identity is maintained independent of SHH.

Examination of in vitro cultures of early postnatal neural progenitors indicated that, at the timepoints examined, neither neuronal nor glial lineages are reproducibly affected by alterations in SHH signaling. However, ablation of GLI3 in the adult dorsal V-SVZ in vivo resulted in decreased abundance of OLIG2-positive progeny, indicating a requirement for regulation of SHH signaling in oligodendrocyte production. These data add to those from prior studies highlighting a role for SHH pathway activation, in which mutant SMOM2 was ectopically activated in dorsal V-SVZ and increased production of oligodendrocytes was observed [8]. However, loss of GLI3 may not allow full activation of this pathway in a manner equivalent to SMOM2 induction. Other studies have indicated a requirement for both activation and subsequent repression (and thus tight regulation) of the SHH response in oligodendrocyte differentiation after demyelinating injury [22]. We hypothesize that in the absence of GLI3, this regulation is disrupted, preventing the normal generation of oligodendrogenic progeny. In future studies, it will be of interest to further dissect this role in the glial lineage, and to determine whether GLI3 regulates the requirement for GLI1 inhibition in the oligodendrogenic response of the demyelinated brain, where remyelination is thought to be driven primarily by ventral V-SVZ cell activity [22,23,40].

## Figures and Tables

**Figure 1 cells-11-00218-f001:**
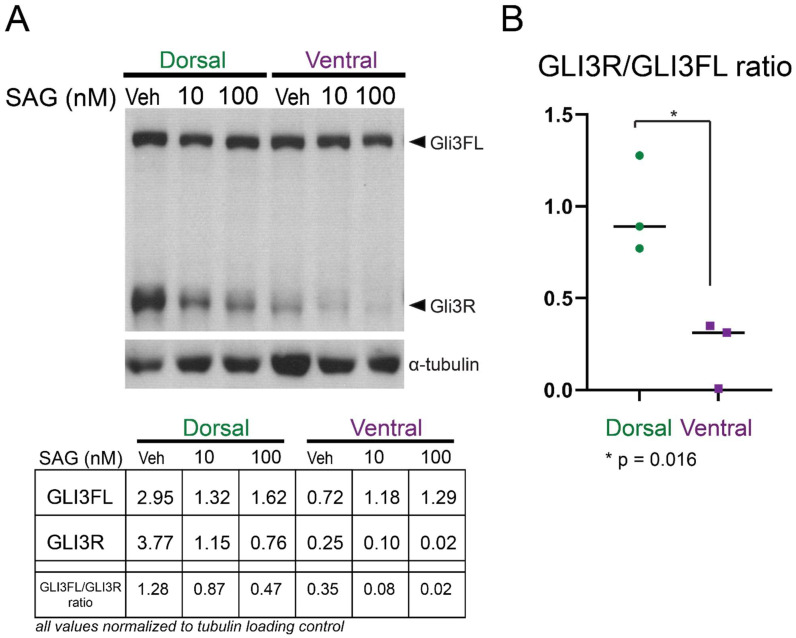
Gli3R is more abundant in cultured cells from the dorsal V-SVZ: (**A**) A representative Western blot of microdissected, cultured progenitors from P2 dorsal and ventral V-SVZ is shown. Cells cultured in N5 media were incubated for 24 h with vehicle or Smoothened agonist (SAG) at a concentration of 10 nM or 100 nM. Dorsal cells incubated with vehicle show abundant GLI3R, which is reduced upon incubation with SAG. Ventral cells also have detectable full-length GLI3 protein (GLI3FL) but reduced amounts of GLI3R. Alpha-tubulin was used as a loading control. Quantification of band intensity for the blot shown is given in the table below. (**B**) The relative amount of GLI3R to GLI3FL is shown; each dot represents an independent biological replicate Western blot analysis. Lines represent the median value for each condition. The relative amount of GLI3R versus GLI3FL is higher in dorsal cells than ventral (*p* = 0.016, unpaired *t*-test).

**Figure 2 cells-11-00218-f002:**
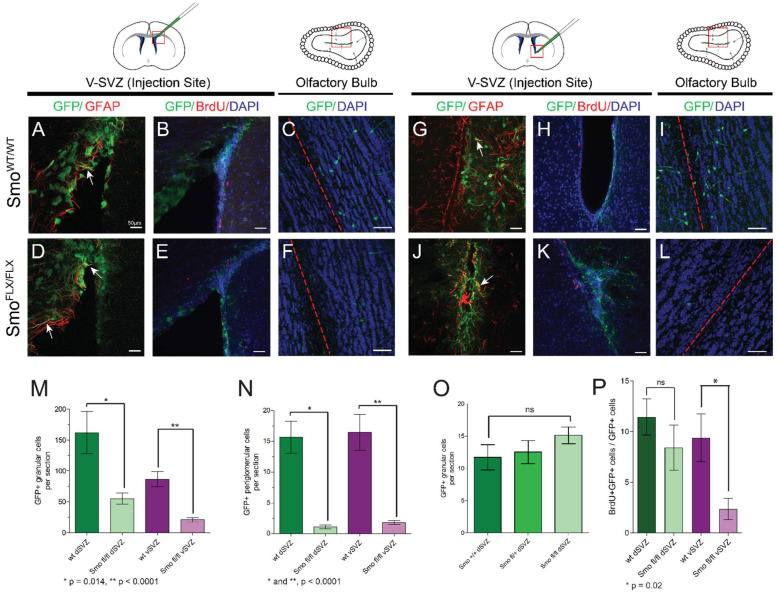
SMO is required for production of olfactory bulb interneurons in V-SVZ: Ad-GFAPp-Cre virus was injected in either dorsal (**A**–**F**) or ventral (**G**–**L**) V-SVZ in CAG or Smo^flx/flx^; CAG animals, and brain tissue was analyzed 30 days after injection. Approximate locations of the imaged areas are indicated by red boxes. GFP/GFAP double-labeled cells were found at the injection site (arrows in **A**,**D**,**G**,**J**), and GFP-labeled progeny were scored in the olfactory bulb for all sites and genotypes (**C**,**F**,**I**,**L**). Scale bars—50 microns; red dashed lines indicate the core of the olfactory bulb. Loss of Smoothened resulted in significant decreases in labeled cells in both the granule cell layer (**M**) and periglomerular cell layer (**N**) of the olfactory bulb, regardless of whether virus was injected dorsally or ventrally. Bar plots each show mean value +/− SEM (*n* = 4 to 5 sections each from each of 4 or 5 animals per location and genotype; *p* values for unpaired *t*-test are shown below each plot). This difference was not seen in control dorsal injections of Ad:CMVp-GFP (**O**) (*n* = 4 to 5 sections each from each of 3 or 4 animals per genotype). (**P**) Quantification of BrdU/GFP double-positive cells at the site of injection showed a decrease in BrdU incorporation in labeled cells of the ventral V-SVZ (*n* = 3 to 4 sections each from each of 3 or 4 animals per genotype; *p* value for unpaired *t*-test is shown beneath plot).

**Figure 3 cells-11-00218-f003:**
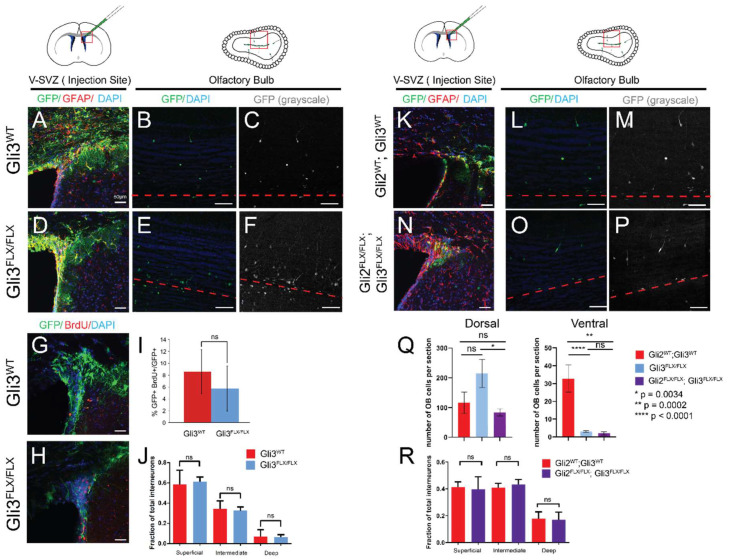
Removal of GLI3 or GLI2/GLI3 does not alter olfactory interneuron production in the dorsal V-SVZ: Ad-GFAPp-Cre virus was injected in dorsal V-SVZ in *CAG*, *Gli3^FLX/FLX^*; *CAG*, or *Gli2^FLX/FLX^*; *Gli3^FLX/FLX^*; *CAG* animals, and brain tissue was analyzed 30 days after injection. Approximate locations of the imaged areas are indicated by red boxes. GFP/GFAP double-labeled cells were found at the injection site (**A**,**D**,**K**,**N**), and GFP-labeled progeny were scored in the olfactory bulb for all sites and genotypes (**B**,**E**,**L**,**O**; GFP alone shown in **C**,**F**,**M**,**P**). Scale bars—50 microns; red dashed lines indicate the core of the olfactory bulb. Ablation of GLI3 in the dorsal V-SVZ did not alter the fraction of GFP-positive cells that also incorporated BrdU (**G**–**I**). No difference in the location of interneurons in the granular layer was seen for any genotype (**J**,**R**). Ventral injection in either *Gli3^FLX/FLX^*; *CAG* animals or *Gli2^FLX/FLX^*; *Gli3^FLX/FLX^*; *CAG* animals resulted in a substantial reduction in the number of interneurons per tissue slice in the olfactory bulb; this was not seen after dorsal injections (**Q**).

**Figure 4 cells-11-00218-f004:**
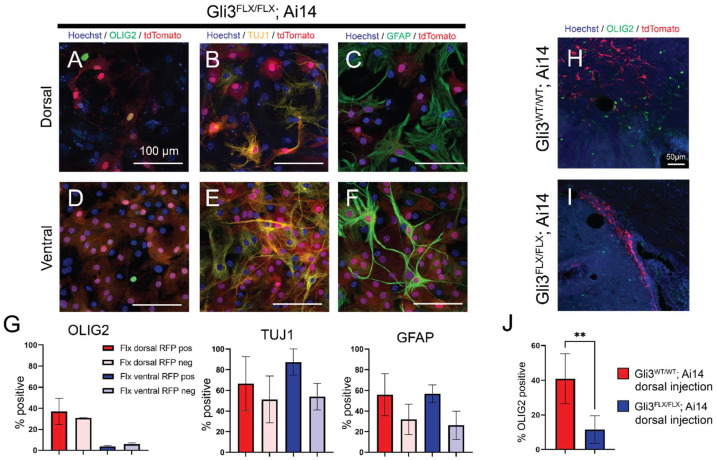
In vitro differentiation is not altered upon ablation of GLI3 in cultured V-SVZ progenitors, but in vivo oligodendrogenesis is altered in adults. Cultures derived from *Gli3^FLX/FLX^*; *Ai14* animals were grown in standard neural stem cell proliferation media and transduced with Ad:CMVp-Cre prior to removal of growth factors and induction of differentiation. After 21 days, cultures were analyzed for the expression of OLIG2 (**A**,**D**), TUJ1 (**B**,**E**), and GFAP (**C**,**F**). Quantification of labeling for each marker in RFP-positive and RFP-negative cells from *Gli3^FLX/FLX^*; *Ai14* cultures is shown in (**G**). Bars represent mean +/− SEM across 3 independent replicates. D/V = dorsal or ventral region. To test the effects of GLI3 ablation in vivo, adult (P30–P40) animals were injected with Ad:GFAPp-Cre and labeled progeny were assessed at 30 days post injection for OLIG2 expression (**H**,**I**). The fraction of RFP-labeled progeny expressing OLIG2 was significantly decreased in *Gli3^FLX/FLX^*; *Ai14* animals (**J**). Bar charts represent mean +/− SEM across 3 (*Ai14*) or 5 (*Gli3^FLX/FLX^*; *Ai14*) independent biological replicates. ** *p* = 0.0089, unpaired *t*-test.

**Table 1 cells-11-00218-t001:** Antibodies used for staining show in detail.

Host/Target Antigen	Clone/Source	Dilution Used
Guinea pig anti-DCX	Millipore AB2253	1:1000
Chicken anti-GFAP	Abcam ab4674	1:2500
Mouse anti-GFAP	Chemicon GA5 MAB3402	1:1000
Chicken anti-GFP	Aves Labs GFP-1020	1:1000
Rabbit anti-GLI3	Genentech 6F5 [31]	1:1000
Rabbit anti-OLIG2	Millipore AB9610	1:500
Rat anti-RFP	Chromotek 5F8	1:1000
Mouse anti-TUJ1	Promega G712A	1:1000

## Data Availability

Original Western blot films and raw imaging data files as well as quantification outputs are available from the corresponding author upon reasonable request.

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
