# Peer review of "GLI3 Is Required for OLIG2+ Progeny Production in Adult Dorsal Neural Stem Cells"

_cells, 2022, doi:10.3390/cells11020218_

Round 1

Reviewer 1 Report

In this manuscript entitled « The GLI3 repressor is enriched in subpopulations of oligodendrogenic postnatal stem cells », the authors investigate the potential role of Hedgehog signaling in the subventricular zone (SVZ) of the lateral ventricles in the postnatal mouse. This germinal niche is sub-divided into dorsal and ventral regions that generate distinct olfactory neurons and provide oligodendrocytes to the corpus callosum. The authors have used stereotaxic injections of Ad:GFAPp-Cre at dorsal and ventral regions of the SVZ in adult mice. They analysed various mouse lines (Smofl/fl, Gli3fl/fl or Gli2fl/fl:Gli3fl:fl) for investigating the deletion of Smoothened, a GPCR implicated in Hedgehog transduction , or the transcription factors Gli2 and Gli3 associated to Hedgehog signaling. CAG and Ai14 lines are used for controlling recombination. They examine the number of GFP+ cells that reaches the olfactory bulbs one month after recombination and investigate the proliferation status of GFP+ cells in the SVZ of these animals. They also provide biochemical and cellular analysis of cultured cells derived from the dorsal and the ventral part of the mouse SVZ at postnatal day 2 in WT animals and after ablation of Gli3.

The described experiments indicate that Smo, from the dorsal and ventral SVZ, is required for the generation of olfactory bulb interneurons. The authors claim that Gli3 repressor (GliR) is enriched in dorsal SVZ cells by Western blot analysis of cultured progenitors from P2 animals. However, removal of Gli3 alone or in combination with Gli2 in dorsal SVZ cells after Ad:GFAPp-Cre injections in mutant animals does not modify the distribution of interneurons in the superficial, intermediate of deep layers of the granular layer. Interestingly, Olig2 positive cells are increased in cultures from dorsal SVZ regions compared to ventral regions. The authors claim that the proportion of these Olig2+ cells in these cultures is increased when Gli3 is abrogated upon recombination with Ad:CMV-Cre.

Altogether, these experiments support that Gli3 activity is not required for the generation of olfactory interneurons from dorsal SVZ progenitors but is rather implicated in the regulation of oligodendrocyte progenitors. These data should be of interest for further delineating the role of dorsal SVZ progenitors in the generation of remyelineating oligodendrocytes in the corpus callosum in vivo.

Major comments :

1- The title of the manuscript is misleading and should be changed : The authors do not show enrichment of GLI3 repressor in a subpopulation of oligodendrogenic postnatal neural stem cells.

There is no evidence that i) GLI3R signal observed from cultured cell extracts from dorsal and ventral P2 SVZ (Figure 1) is enriched in oligodendrogenic cells, ii) that. Gli3/GLI3 distribution occurs only in Olig2+ (and not in Tuj1+ or GFAP+) in P2 SVZ cultured cells.

2 - Page 8 : The authors claim : « A higher fraction of OLIG2+ cells were seen in dorsally derived control cultures (Figures 4 A,D, G, J, M) relative to matched ventral counterparts ».

In this figure, analysis of mean intensity per cell should be also provided for RFP+OLIG2+, RFP+Tuj1+, RFP+GFAP+ cells and RFP-OLIG2+, RFP-Tuj1+, RFP-GFAP+ cells from dorsal and ventral Gli3WT ; Ai14 cultures.

3- Page 8 : The authors claim : « Interestingly, the proportion of cells which were OLIG2+further increased when GLI3 was lost »

The authors should provide the analysis showing this claim.

Comments :

1- In Methods, statistical analysis used should be indicated in the text when required.

2- Specificity of the Genentech 95.9 rabbit anti-Gli3 antibody used in Figure 1 towards Gli1 and Gli2 should be specified/discussed.

3- Please, indicate what are the data shown in Figures 2I-L and Figure 3G, H, and O. (mean+/- SEM ?)

4- Please, specify how many independent cultures have been analyzed by Western blot for Gli3 expression.

5- In Results, page 5, last sentence of the first paragragh : Please, indicate the concentration of cyclopamine used in the cultures.

6- Figure 2K, Please, specify the number of animals used in this experiment.

7- Figures 2 A, C, E, G : Please, indicate in the figures the location of GFP/GFAP double-labeled cells as claimed in the legend.

8- Figure 2K. Sections showing double GFP/BrdU positive cells would improve the manuscript.

9- Page 6, first sentence : « Given the elevated levels of Gli3 in the dorsal portion of the niche, we next … ». This sentence should specify Gli3R (and not Gli3) and indicate that these data are from P2 cultured cells.

10- Page 7, Figure 3 : Please, specify what are the signals shown in C, F, K and N.

In this manuscript entitled « The GLI3 repressor is enriched in subpopulations of oligodendrogenic postnatal stem cells », the authors investigate the potential role of Hedgehog signaling in the subventricular zone (SVZ) of the lateral ventricles in the postnatal mouse. This germinal niche is sub-divided into dorsal and ventral regions that generate distinct olfactory neurons and provide oligodendrocytes to the corpus callosum. The authors have used stereotaxic injections of Ad:GFAPp-Cre at dorsal and ventral regions of the SVZ in adult mice. They analysed various mouse lines (Smofl/fl, Gli3fl/fl or Gli2fl/fl:Gli3fl:fl) for investigating the deletion of Smoothened, a GPCR implicated in Hedgehog transduction , or the transcription factors Gli2 and Gli3 associated to Hedgehog signaling. CAG and Ai14 lines are used for controlling recombination. They examine the number of GFP+ cells that reaches the olfactory bulbs one month after recombination and investigate the proliferation status of GFP+ cells in the SVZ of these animals. They also provide biochemical and cellular analysis of cultured cells derived from the dorsal and the ventral part of the mouse SVZ at postnatal day 2 in WT animals and after ablation of Gli3.

The described experiments indicate that Smo, from the dorsal and ventral SVZ, is required for the generation of olfactory bulb interneurons. The authors claim that Gli3 repressor (GliR) is enriched in dorsal SVZ cells by Western blot analysis of cultured progenitors from P2 animals. However, removal of Gli3 alone or in combination with Gli2 in dorsal SVZ cells after Ad:GFAPp-Cre injections in mutant animals does not modify the distribution of interneurons in the superficial, intermediate of deep layers of the granular layer. Interestingly, Olig2 positive cells are increased in cultures from dorsal SVZ regions compared to ventral regions. The authors claim that the proportion of these Olig2+ cells in these cultures is increased when Gli3 is abrogated upon recombination with Ad:CMV-Cre.

Altogether, these experiments support that Gli3 activity is not required for the generation of olfactory interneurons from dorsal SVZ progenitors but is rather implicated in the regulation of oligodendrocyte progenitors. These data should be of interest for further delineating the role of dorsal SVZ progenitors in the generation of remyelineating oligodendrocytes in the corpus callosum in vivo.

Major comments :

1- The title of the manuscript is misleading and should be changed : The authors do not show enrichment of GLI3 repressor in a subpopulation of oligodendrogenic postnatal neural stem cells.

There is no evidence that i) GLI3R signal observed from cultured cell extracts from dorsal and ventral P2 SVZ (Figure 1) is enriched in oligodendrogenic cells, ii) that. Gli3/GLI3 distribution occurs only in Olig2+ (and not in Tuj1+ or GFAP+) in P2 SVZ cultured cells.

2 - Page 8 : The authors claim : « A higher fraction of OLIG2+ cells were seen in dorsally derived control cultures (Figures 4 A,D, G, J, M) relative to matched ventral counterparts ».

In this figure, analysis of mean intensity per cell should be also provided for RFP+OLIG2+, RFP+Tuj1+, RFP+GFAP+ cells and RFP-OLIG2+, RFP-Tuj1+, RFP-GFAP+ cells from dorsal and ventral Gli3WT ; Ai14 cultures.

3- Page 8 : The authors claim : « Interestingly, the proportion of cells which were OLIG2+further increased when GLI3 was lost »

The authors should provide the analysis showing this claim.

Comments :

1- In Methods, statistical analysis used should be indicated in the text when required.

2- Specificity of the Genentech 95.9 rabbit anti-Gli3 antibody used in Figure 1 towards Gli1 and Gli2 should be specified/discussed.

3- Please, indicate what are the data shown in Figures 2I-L and Figure 3G, H, and O. (mean+/- SEM ?)

4- Please, specify how many independent cultures have been analyzed by Western blot for Gli3 expression.

5- In Results, page 5, last sentence of the first paragragh : Please, indicate the concentration of cyclopamine used in the cultures.

6- Figure 2K, Please, specify the number of animals used in this experiment.

7- Figures 2 A, C, E, G : Please, indicate in the figures the location of GFP/GFAP double-labeled cells as claimed in the legend.

8- Figure 2K. Sections showing double GFP/BrdU positive cells would improve the manuscript.

9- Page 6, first sentence : « Given the elevated levels of Gli3 in the dorsal portion of the niche, we next … ». This sentence should specify Gli3R (and not Gli3) and indicate that these data are from P2 cultured cells.

10- Page 7, Figure 3 : Please, specify what are the signals shown in C, F, K and N.

Reviewer 2 Report

This is a brief report examining the role of Gli3 in the V-SVZ neural stem cells. This study found that the active form of Gli3 (Gli3FL) is expressed in cultured cells from both dorsal and ventral SVZ of P2 mice, using western blot. Stereotactic injections of adeno-Cre viruses were then used to ablate Smo, Gli2 and/or Gli3 in the dorsal vs. ventral SVZ of adult (P30-60) mice. Ablation of Smo reduced the granular and periglomerular cells derived from both ventral and dorsal SVZ. However, proliferation was only reduced in the ventral cells with loss of Smo. In contrast, Gli3 ablation neither altered the number/location of olfactory bulb interneurons nor the proliferation of SVZ cells. Similarly, combined loss of Gli2 and Gli3 did not change the distribution of interneurons in the olfactory bulb. Next, the effect of loss of Gli3 on differentiation of dorsal vs. ventral SVZ cells from P2 mouse brains was examined in vitro. Overall the dorsal derived cells generated more oligodendrocytes and less neurons compared to ventral derived cells. The Gli3-deficient dorsal cells generated a higher fraction of the oligodendrocytes and astrocytes; the Gli3-deficient ventral cells in contrast generated less neurons compared to the controls expressing Gli3. While the role of Gli3 in oligodendrogenesis is a novel finding, there are several weaknesses in the experimental design that render the results inconclusive.  

  1. The authors performed all the in vitro studies in cells from P2 mice while the in vivo studies were performed in adult (P30-60) mice. Developmental myelination occurs post-natally in the mammalian brain, predominantly within the first 4 weeks. Other studies have already shown that ventral cells from P6-9 mice can generate oligodendrocytes in vivo but fail to do so in the adult brain (PMID: 29045809 and PMID: 26416758). So, to compare the results from the cell culture experiments with the in vivo results, the in vitro studies should be performed with cells from the adult SVZ.
  1. Figure 1: The expression of Gli3FL vs. Gli3R should be examined in the adult (P30-60) SVZ. The western blot data in Fig.1 should be quantified and normalized to the loading control since the levels of loading control are not the same in every lane.
  2. Figure 2: The recombination efficiency of the Smo-FL alleles should be confirmed. An inset showing a higher magnification view of GFP+GFAP cell would help the reader.
  3. Figure 3: The recombination efficiency of the Gli2-FL and Gli3-FL alleles should be confirmed.
  4. The convention for floxed alleles should be changed to FX instead of FL to avoid confusion with Gli3-FL in figure 1.
  5. The effect of combined loss of Gli2 and Gli3 on proliferation should be examined especially since loss of smo produced a decrease in ventral cells.
  6. Figure 4: This experiment should be done with cells from the adult brain. Several controls are missing from this experiment. Here, the uninfected cells within the same well were used as controls. To rule out preferential infection of a subset of cells and effect of infection on the cells, cells from the same genotype should be infected with the virus expressing only GFP and cells from Ai14 only mice should be infected with the Cre virus.
  7. Figure 4: Scale is not mentioned in the legend. The images are not clear. An inset with higher mag view would help. The data have not been statistically analyzed. The positive fraction of ventral cells as well as negative fraction of dorsal and ventral cells do not add up to 100%. The data should be plotted graphically to show the differences between the groups.
  8. Statistical methods have not been mentioned throughout the manuscript.
  9. The sex of the mice used in every experiment should be mentioned whenever possible.

Round 2

Reviewer 1 Report

The manucript has been improved.

Author Response

We thank the reviewer for their suggestions and are pleased that they feel the manuscript is improved.

Reviewer 2 Report

The authors have addressed my comments adequately although I would have liked to see the in vitro experiments repeated with adult neural stem cells. The finding that loss of Gli3 in dorsal SVZ in vivo reduces Olig2+ cells is very interesting. However, whether these Olig2+ cells differentiate into oligodendrocytes is still unknown since Olig2+ cells can differentiate into astrocytes as well.  Hence the title is misleading since oligodendrocyte production has not been confirmed with markers of mature oligodendrocytes. 

Author Response

We take the reviewer's point regarding the title and suggest the following revised title: "GLI3 is required for OLIG2+ progeny production in adult dorsal neural stem cells."